# Depletion of Alpha-Melanocyte-Stimulating Hormone Induces Insatiable Appetite and Gains in Energy Reserves and Body Weight in Zebrafish

**DOI:** 10.3390/biomedicines9080941

**Published:** 2021-08-02

**Authors:** Yang-Wen Hsieh, Yi-Wen Tsai, Hsin-Hung Lai, Chi-Yu Lai, Chiu-Ya Lin, Guor Mour Her

**Affiliations:** 1Department of Bioscience and Biotechnology, National Taiwan Ocean University, Keelung 202, Taiwan; hearhero@hotmail.com (Y.-W.H.); c.y.stephen.lai@gmail.com (C.-Y.L.); vista_jckey_1590@livemail.tw (C.-Y.L.); 2Institute of Biopharmaceutical Sciences, National Yang Ming Chiao Tung University, Taipei 112, Taiwan; s232579@gmail.com; 3Department of Family Medicine, Chang Gung Memorial Hospital, Keelung 204, Taiwan; tsaiyiwen@gmail.com; 4College of Medicine, Chang Gung University, Taoyuan 333, Taiwan

**Keywords:** orexigen, obesogens, adipogenesis, hypothalamus, appetite

## Abstract

The functions of anorexigenic neurons secreting proopiomelanocortin (POMC)/alpha-melanocyte-stimulating hormone (α-MSH) of the melanocortin system in the hypothalamus in vertebrates are energy homeostasis, food intake, and body weight regulation. However, the mechanisms remain elusive. This article reports on zebrafish that have been genetically engineered to produce α-MSH mutants, α-MSH^−7aa^ and α-MSH^−8aa^, selectively lacking 7 and 8 amino acids within the α-MSH region, but retaining most of the other normal melanocortin-signaling (Pomc-derived) peptides. The α-MSH mutants exhibited hyperphagic phenotypes leading to body weight gain, as observed in human patients and mammalian models. The actions of several genes regulating appetite in zebrafish are similar to those in mammals when analyzed using gene expression analysis. These include four selected orexigenic genes: Promelanin-concentrating hormone (*pmch*), agouti-related protein 2 (*agrp2*), neuropeptide Y (*npy*), and hypothalamic hypocretin/orexin (*hcrt*). We also study five selected anorexigenic genes: Brain-derived neurotrophic factor (*bdnf*), single-minded homolog 1-a (*sim1a*), corticotropin-releasing hormone b (*crhb*), thyrotropin-releasing hormone (*trh*), and prohormone convertase 2 (*pcsk2*). The orexigenic actions of α-MSH mutants are rescued completely after hindbrain ventricle injection with a synthetic analog of α-MSH and a melanocortin receptor agonist, Melanotan II. We evaluate the adverse effects of MSH depletion on energy balance using the Alamar Blue metabolic rate assay. Our results show that α-MSH is a key regulator of POMC signaling in appetite regulation and energy expenditure, suggesting that it might be a potential therapeutic target for treating human obesity.

## 1. Introduction

The arcuate melanocortin system in vertebrates consists of anorexigenic neurons expressing proopiomelanocortin peptide (POMC), and orexigenic neurons expressing neuropeptide Y/agouti-related peptide (NPY/AgRP). The POMC-expressing neurons of the hypothalamic arcuate nucleus (ARC) participate in the control of food intake, energy homeostasis, body weight (BW) regulation, and other metabolic processes in the melanocortin system [1]. The POMC precursor peptide is processed into a series of biologically active components, including alpha-melanocyte-stimulating hormone (α-MSH), adrenocorticotrophic hormone (ACTH), β-MSH, and β-endorphin (β-END) by tissue-specific proteolysis [2,3]. α-MSH functions in anorexigenic responses by activating the melanocortin 4 receptor (MC4R), which is expressed on distinct second-order neurons, whereas AGRP, the orexigenic peptide, conveys as an antagonist of MC4R [4,5]. In mammals, loss of the genes encoding POMC [6] or MC4R [7,8] leads to severe obesity.

Melanocortin peptides are known to regulate feeding behavior in mammals [9,10]. The suppression of appetite in mammals mediated by α-MSH/β-MSH is attributed to MC4R [7]. In animal models, hyperphagia, an obese phenotype, and hyperinsulinemia were observed in the obese yellow mouse (Ay) model of interruption of α-MSH central signaling by the ubiquitous constitutive expression of the agouti gene [11]. Recently, genetically modified *Pomc^tm1/tm1^* mice, which had a mutation in the *Pomc* gene, were unable to synthesize desacetyl-α-MSH and α-MSH. *Pomc^tm1/tm1^* mice are hyperphagic and show an obese phenotype even when fed a regular chow diet [12,13]. It was recently reported in a model using Labrador retriever dogs that a 14-bp deletion in the gene encoding pro-POMC in these canines is associated with obesity [14]. The β-MSH [15,16,17] and β-endorphin [18,19] coding sequences are also functionally associated with adiposity and greater appetite [20]. In fact, novel mutations (Phe144Leu [21] and Arg145Cys [22]) located in the α-MSH domain of the *POMC* gene were observed to be associated with early-onset obesity.

Additionally, moderately linear growth, which is primarily regulated by growth hormone (GH) released by somatotrophs in the adenohypophysis of the pituitary gland, was also observed in both loss-of-function mutant *Pomc* or *Mc4r* rodents and human models [6,7,23,24,25,26]. Familial glucocorticoid deficiency (FGD), whose clinical features are enhanced longitudinal bone growth and advanced bone age, is an ACTH-insensitivity disorder characterized by the overproduction of ACTH [27,28,29].

According to previous studies, Pomc in zebrafish has highly conserved regions similar to ACTH, α-MSH, β-END, β-MSH, and possibly to N-POMC-derived peptides, but lacks γ-MSH compared with mammalian models [30]. However, among the two *pomc* family member genes, *pomca* and *pomcb*, observed in zebrafish [2], only *pomca*, expressed in the pituitary gland, is responsible for developing the interrenal organ of zebrafish [30,31]. In zebrafish, knockdown of the gene encoding Pomca resulted in a significant reduction in Acth immunoreactivity, and attenuated melanosome dispersal at five days postfertilization (dpf) following injection of a designed antisense morpholino oligonucleotide [32]. Shi et al. demonstrated that the increased somatic growth without obesity in *pomca* mutant zebrafish was associated with reduced anxiety-like behaviors but not with appetite or energy expenditure.

Most research on the biological functions of Pomc in zebrafish focusing on anxiety disorder has been based on the evaluation of changes in Acth expression levels. Here, we used zebrafish to test whether depletion of α-MSH signaling might disturb food intake and influence obesity, as the hypothalamic neural circuits involved in food intake are highly conserved in fish species. Our findings provide comprehensive information about the dynamic expression of genes controlling appetite and growth in nonfunctional α-MSH signaling in the Pomc neurons of zebrafish.

## 2. Materials and Methods

### 2.1. Fish Husbandry

All zebrafish were maintained in a controlled environment with a 14/10-h light-dark cycle at 28 °C. They were fed twice daily with brine shrimp and commercial fish food pellets. All animal experiments were conducted in accordance with the guidelines and approval of the Institutional Animal Care and Use Committee (IACUC) of the National Yang Ming Chiao Tung University, Taiwan.

### 2.2. TALEN Cloning and Targeted Mutagenesis

For each mutant target site in the *pomca* locus(acc. no. NM_181438.3), two 18 bp TALEN binding sites were selected (exon3: 5′-CCTACTCCATGGAGCAC-3′, 5′-CGGTCGGCCGCAAACGC-3′). A restriction enzyme (AgeI) site between each TALEN pair was used for genotyping by restriction fragment length polymorphism (RFLP) analysis. TALEN constructs were cloned using Golden Gate assembly (Cermak et al., 2011) and an accompanying plasmid kit from Addgene (Addgene Kit #1000000024). TALEN mRNAs were injected into 1-cell embryos to generate stable mutant lines. The final TALEN expression plasmids were linearized by digestion with the NotI restriction enzyme. TALEN mRNAs were transcribed using the mMESSAGE SP6 kit (Ambion, USA) and purified using the RNeasy Mini kit (QIAGEN, Hilden, Germany).

### 2.3. Quantitative Reverse Transcription Polymerase Chain Reaction (RT–qPCR)

Total RNA was extracted using TRIzol reagent (Thermo Fisher Scientific, USA) or the RNeasy Mini Kit (QIAGEN, Hilden, Germany) and reverse transcribed using the first-strand cDNA synthesis kit (K1691; Thermo Fisher Scientific). Real-time RT–qPCR was performed using the StepOne Real-Time PCR System (Applied Biosystems, Foster City, USA). The genes and their corresponding primer sequences are listed in Appendix A. These genes, including those encoding growth hormone (*gh*), growth hormone receptor b (*ghrb*), insulin-like growth factor 2b (*igf2b*), insulin-like growth factor binding protein 1b (*igfbp1b*), insulin-like growth factor binding protein 5b (igfbp5b), and insulin-like growth factor binding protein 6b (*igfbp6b*) have been validated in zebrafish [33].

### 2.4. In-Situ Hybridization (ISH)

The gene-specific probes were cloned by PCR into a pGEM^®^-T Easy TA cloning vector (Promega, Madison, WI USA) using the primers listed in Appendix A. Antisense probes were synthesized by in vitro transcription using the DIG RNA Labelling Kit (SP6/T7) (Roche Applied Science, Mannheim, Germany). ISH was performed as described previously [34].

### 2.5. Quantification of Food Intake

Our procedure was adapted from that described previously [35]. The larval fish food was fluorescently labeled paramecia prepared using the lipophilic tracer 4-(4-(Didecylamino)styryl)-*N*-Dethylpyridinium iodide (4-Di-10-ASP; Invitrogen, Carlsbad, CA, USA). We conducted feeding of 7 dpf zebrafish in 6-well plates to allow for free swimming. At 1.5 h after feeding, the larvae were anesthetized. After two washes to remove residual paramecia, groups of five larvae were transferred into a 96-well round-bottom black plate in an anesthetic solution. The intra-abdominal fluorescent signal was measured using the Synergy™ HT Multi-Detection Reader (BioTek Instruments, Winooski, VT) in fluorescence area scan mode 11 × 11 multipoint/well, 0.1 s/point using a fluorescein filter set (excitation wavelength, 485 nm; emission wavelength, 590 nm).

### 2.6. Hindbrain Ventricle Injection of Zebrafish Larvae

Briefly, anesthetized fish were placed on a 3% agarose plate with water containing 0.05% MS222. The skulls were impaled with a 0.53 mm diameter needle attached to a syringe in the midline at the telencephalon–diencephalon border. The fish were injected intracranially with 4.6 nL of sterile water, an α-MSH analog (100 μM in sterile water; M4135, Sigma-Aldrich), or Melanotan II (MTII, 100 μM in sterile water; M8693, Sigma-Aldrich) into the cranial cavity by a heat-pulled glass capillary micropipette attached to microinjection equipment. After administration, the cut heads were sampled, and at least 30 larvae were pooled to yield enough total RNA.

### 2.7. Histology and Immunohistochemistry (IHC)

Tissues were fixed using 4% paraformaldehyde solution in phosphate-buffered saline (PBS) overnight at 4 °C, washed in PBS, and equilibrated in 30% sucrose/PBS overnight at 4 °C. Then, they were embedded in an OCT compound and cut into 4 μm sections using a cryostat. IHC was performed as described previously [36,37]. Rabbit anti-α-MSH antibody (1:1000, RayBiotech 130-10355) was used to detect α-MSH immunoreactivity. Protein levels were detected using horseradish peroxidase (HRP)-conjugated secondary antibody (1:1000, Jackson Immuno Research AB_2313567).

### 2.8. Alamar Blue Metabolic Rate Assay

Our metabolic rate assay was adapted from an established method [38]. The assay buffer was supplemented with egg water containing 0.1% DMSO, 1% Alamar Blue (A50101, Thermo Fisher Scientific), and 4 mM sodium bicarbonate. Zebrafish were rinsed with 0.22 μm filtered egg water and pipetted into a 96-well plate (three specimens per well). Fluorescence of the plate was immediately read on the Infinite 200 PRO multimode plate reader (Tecan Group Ltd., Switzerland) with excitation at 530 nm and emission monitored at 590 nm (fluorescence area scan mode 3 × 3 multipoint/well, 10 times/point per reading). The plate was incubated in the dark at 28 °C for 24 h and then read again. Any wells containing dead fish were excluded from the analyses. The change in fluorescence from time 0 to 24 h was then calculated. The data were corrected by setting the average of the wild-type (WT) controls to 1.

### 2.9. Whole-Mount Oil Red O Staining

Zebrafish larvae were fixed in 4% paraformaldehyde solution in PBS overnight at 4 °C. Equal numbers of control and mutant larvae were transferred to 1.5 mL Eppendorf tubes and rinsed three times (5 min each) with PBS. After removing the PBS, the larvae were prestained with a mixture of 60% isopropyl alcohol for at least 1 h. Then, fresh 0.5% Oil Red O solution was added at 4 °C for 1 h, and larvae were washed in PBS. They were stored in 70% glycerol and imaged using a bright-field dissecting microscope (Stemi 305, Carl Zeiss AG, Oberkochen, Germany).

### 2.10. Growth Rate

The growth rate was recorded monthly starting at two months postfertilization (mpf) and ending at 12 mpf. Twenty fish per diet (mixed sex) were selected randomly. Body length (BL) was measured using a standard metric ruler and determined the distance from the snout to the caudal peduncle. BL (cm) and BW (mg) were used to calculate the body mass index (BMI).

### 2.11. Morphological and Morphometric Studies: Analysis of Zebrafish Fat Tissues and Adipocytes

These studies were carried out on histological sections according to Mon-Talbano et al. [39].

### 2.12. Statistical Analysis

All data are presented as the mean ± standard error of the mean (SEM). Statistical analysis was performed using analysis of variance (ANOVA) followed by Bonferroni post-hoc tests. All analyses were performed using GraphPad Prism software (v. 8.0; GraphPad, San Diego, CA, USA). Differences were considered statistically significant at *p* < 0.05.

## 3. Results

### 3.1. Generation of α-MSH Depletion Lines in Zebrafish

The TALEN-based genome editing technique was performed for the depletion of α-MSH. According to the sequence information, the targeted site of TALENs was located in the third exon of the *pomca* gene locus. Both arms of the designed binding arms were 18 bp long. The spacer between the two arms was 17 bp. The AgeI restriction digestion site within the space region was used for genotyping (Figure 1A).

Three *pomca* mutants were generated here. Two independent α-MSH-depleted mutant lines, α-MSH^−7aa^ and α-MSH^−8aa^ were generated with 21-bp and 24-bp bp in-frame deletions, respectively. One independent *pomca* mutant line, POMCa^141a^, was generated with a 13-bp frameshift deletion (Figure 1A,B). The α-MSH^−7aa^ and α-MSH^−8aa^ mutations resulted in in-frame deletions that produced truncated proteins of 7 amino acids (aa) and 8 aa within the α-MSH regions, respectively (Figure 1C,D). No significantly decreased expression pattern of *pomca* transcripts was observed in the pituitary gland or hypothalamus of the α-MSH mutants, except for the POMCa^141a^ mutant compared with the WT controls at 5 dpf (Figure 1E). The α-MSH mutants caused deficiencies of only the α-MSH peptide hormones (Figure 1F), whereas the POMCa^141a^ mutant resulted in premature stop codons that produced truncated proteins of 141 aa (Figure 1C,D), which caused a deficiency in the majority of melanocortin peptides derived from POMCa (Figure 1F). In addition, three *pomca* mutants showed attenuated melanosome dispersion in dark conditions, and a significant decrease in melanosomes was observed in the POMCa^141a^ mutant (Figure 1G). 

### 3.2. Defective α-MSH Increased Food Intake in Zebrafish

To evaluate the pattern of food intake at the larval stage of the zebrafish mutants, we used a previously described feeding protocol to prepare fluorescent food composed of larval fish food with lipophilic tracer 4-Di-10-ASP-labeled paramecia [35] (see Methods). The larvae ingested this fluorescent food, which led to an accumulation of fluorescent signals within the intestine that could be visualized readily by fluorescence microscopy (Figure 2A).

The α-MSH^−7aa^ and α-MSH^−8aa^ larvae exhibited obvious hyperphagia at 7 dpf (Figure 2B), when food intake significantly increased by 2.28-and 2.03-fold, respectively, compared with control WT zebrafish (Figure 2C). However, no obvious differences between POMCa^141a^ and WT controls were observed in terms of food intake at 7 dpf (data not shown). 

### 3.3. Defective α-MSH Enhanced Somatic Growth and Decreased Energy Expenditure Concomitant with Liver Steatosis in Zebrafish Larvae

Next, we tested whether growth and energy budget were proportional to food intake in the POMC mutants, and simultaneous observation was necessary to estimate these parameters. Predictably, the α-MSH mutants significantly increased in BL at 8 dpf (Figure 3A). Somatic growth is predominantly controlled by regulating the GH/insulin-like growth factor (IGF) axis in fish [40,41]. Increased mRNA expression of genes controlling the GH/IGF axis accompanied by increased BL was detected in the *pomc* mutants compared with WT controls at 8 dpf (Figure 3B). Using whole-mount ISH analysis, a significantly increased expression pattern of gh was observed in *pomc* mutants compared with WT at 5 dpf (Figure 3C). Thus, these results indicate that the observed growth activation could be attributed to the upregulation of the GH/IGF axis.

To assess the potential roles of α-MSH in the metabolism of the *pomc* mutants for further studies of energy expenditure that might influence obesity, BW, and appetite regulation, we performed the Alamar Blue metabolic rate assay in larvae at 6–8 dpf. The assay showed increased signals of the zebrafish mutants with incubation time compared with the WT controls (Figure 3D). All *pomc* mutants larvae exhibited significantly lower metabolic rates, with 53% to 84% between the 6 and 7 dpf stages. Both the α-MSH^−7aa^ and α-MSH^−8aa^ larvae exhibited lower metabolic rates, with decreases of 84% and 81%, respectively, compared with control WT zebrafish (Figure 3D). However, the POMCa^141a^ larvae exhibited a slightly higher metabolic rate, with an increase of 37% compared with control WT zebrafish at 8 dpf (Figure 3D). Furthermore, hepatic steatosis in the α-MSH mutants increased markedly compared with that in the WT larvae (Figure 3E), thus confirming that lipid accumulation as an energy reserve is reflected in hepatic steatosis.

### 3.4. α-MSH Mutant Adults Develop Characteristic Melanocortin-Related Obesity

Homozygous α-MSH mutant larvae developed normally and were slightly larger than the WT larvae (Figure 3A,E). Surprisingly, by measuring the growth rate of POMC mutants and WT zebrafish, a dramatic increase in BL (Figure 4A,B) and BW (Figure 4C) was detected in both α-MSH mutant adults compared with WT adults 12 months post-fertilization (mpf). The POMCa^141a^ adults showed the same BL as WT adults after 12 mpf and a slight increase in BW compared with WT adults until 9 mpf. (Figure 4C). There were also statistically significant increases in the BMI at 10 mpf for the α-MSH mutant adults (Figure 4D).

The BMI showed the same trend observed for BL and BW (Figure 4D). Examination of the BW data showed a significant increase in viscera and visceral fat contents in both the α-MSH^−7aa^ and α-MSH^−8aa^ groups at the 12 mpf stage (Figure 4E). Morphometric analysis of adipose tissues showed a dramatic difference in development between the α-MSH mutant adults and the WT controls. Both the α-MSH^−7aa^ and α-MSH^−8aa^ adult fish had larger amounts of adipose tissue, with increases of 8- and 12-fold, respectively, compared with control WT zebrafish (Figure 4F,G). Indeed, the average area of both visceral and subcutaneous adipose tissue showed a significant increase in α-MSH mutant adults compared with WT controls, indicating that an increase in food intake and decreased energy expenditure determined the growth of an abundant layer of adipose tissue, resulting in obesity.

### 3.5. Pre- and Postprandial Expression of Appetite-Related Genes in α-MSH Mutants

As several neuronal peptides are involved in regulating food intake and energy balance in zebrafish, as in mammals [42,43,44], appetite-related gene expression levels were investigated in whole α-MSH mutant brains using RT–qPCR. To compare the expression pattern of appetite-related genes between the WT and the *pomca* mutants, we examined several important factors, including four selected orexigenic genes: Promelanin-concentrating hormone (*pmch*), agouti-related protein 2 (*agrp2*), neuropeptide Y (*npy*), and hypothalamic hypocretin/orexin (*hcrt*) (Figure 5A). We also measured the expression levels of five selected anorexigenic genes: Brain-derived neurotrophic factor (*bdnf*), single-minded homolog 1-a (*sim1a*), corticotropin-releasing hormone b (*crhb*), thyrotropin-releasing hormone (*trh*), and prohormone convertase 2 (*pcsk2*) (Figure 5B). Although the analyses showed that the orexigenic mRNA levels were slightly downregulated in POMC mutants compared with those in WT controls at 7 dpf (Figure 5A), the anorexigenic mRNA levels were dramatically downregulated in *pomca* mutants compared with those in WT controls at 7 dpf (Figure 5B). Using whole-mount ISH analysis, significantly decreased expression patterns of *bdnf* and *sim1a* were observed in *pomca* mutants compared with WT at 5 dpf (Figure 5C). In addition, the intestinal orexigenic gene, *ghrelin* (*ghrl*), was dramatically upregulated in *pomca* mutants compared with that in WT adults (Figure 5D). Thus, functional α-MSH signaling is clearly required for transcriptional suppression to some extent of anorexigenic and orexigenic genes and interrupts the balance of hunger and satiety, which in turn possibly increases appetite under normal feeding conditions.

### 3.6. Administration of a Synthetic α-MSH Analog Rescued Hyperphagic Phenotypes in α-MSH Mutants

We investigated whether hindbrain ventricle injections of an α-MSH analog and MTII into α-MSH mutant larvae could reverse the hyperphagic phenotypes (Figure 6A). The feeding evaluation results after the administration of α-MSH analog (Figure 6B, left) and MTII (Figure 6C, left) were obvious in both the α-MSH^−7aa^ and α-MSH^−8aa^ specimens at 7dpf, respectively. Both α-MSH^−7aa^ and α-MSH^−8aa^ fish could be rescued by the α-MSH analog and MTII—similar to normal phenotypes—although their anorexigenic effects were stronger than in WT controls (Figure 6B,C).

To clarify whether the appetite-related gene expression levels were modulated directly by administration of the α-MSH analog in α-MSH mutant larvae, the mRNA levels of the orexigenic genes *npy*, *agrp2*, and *hcrt* were significantly lower than the mRNA levels of the anorexigenic genes *bdnf*, *sim1a*, *chrb*, and *trh* at 1.5 h after intracranial administration compared with the α-MSH mutant larvae (Figure 6D,E).

## 4. Discussion

MSHs are well-known feeding and camouflage behavior-regulated hormones in mammals (Yaswen et al., 1999; Raffan et al., 2016). MC4R is a key gene controlling the α-MSH-mediated suppression of appetite in mammals [4,7,15,17,45]. Intracerebroventricular (icv) or intraperitoneal (ip) injection of MC4R agonists can suppress food intake in goldfish [46], zebrafish [47], spotted sea bass [48], and coho salmon [49], while transgenic zebrafish overexpressing *agrp* demonstrate adipocyte hypertrophy and increased linear growth, leading to an obese phenotype [50]. Additionally, in a zebrafish model, Mrap2a/Mrap2b can not only interact with Mc4r to regulate appetite, and growth patterns, but also help to modulate the sensitivity of Mc4r to α-MSH [47,51]. Increased food intake has also been observed in *mc4r*-deficient zebrafish [52].

The depletion of α-MSH in *Pomc*-null mice resulted in obesity, whereas the blockade of neuronal melanocortin signaling resulted in a decrease in response to centrally administered leptin [53,54]. Moreover, the central administration of α-MSH suppressed food intake and reduced BW gain in rodent models [55,56]. A deficiency of α-MSH in a *Pomc*^tm1/tm1^ mouse produced exacerbated hyperphagia and obesity when feeding with a high-fat diet (HFD) [12]. Our results—consistent with mouse models—showed significant increases in food intake in the α-MSH mutants (Figure 2B,C). Because no α-MSH peptide was produced in α-MSH mutant fish, a significant increase in food intake was found in their larval stages (Figure 2B,C). Increased BW and obesity in adult α-MSH mutants were also developed under regular feeding and overfeeding conditions (Figure 4A,C). However, a zebrafish *pomc* mutant, POMCa^141a^, which was also α-MSH deficient, showed enhanced somatic growth without increasing food intake. This suggests that some aspects in addition to regulating appetite could be the key cause(s) for BW gain in our zebrafish model of POMCa^141a^.

Hyperphagia alone did not promote fish growth. In medaka and zebrafish models, leptin-receptor deficiency also elicited an increased food intake pattern with a normal growth rate [57,58]. Because the energy homeostasis controlling system is highly complicated, simply increasing food intake might not explain the causal relationship with an acceleration of the growth rate or with weight gain [59,60,61]. In the Alamar Blue metabolic assay, we observed that the α-MSH mutant decreased energy expenditure by decreasing its metabolic rate (Figure 3D). Thus, the deletion of α-MSH in zebrafish induced hyperphagia and lower metabolism levels, which resulted in obesity. We found not only increases in food intake but also increases in the growth rate, including the standard BL, BW, and obesity, in the α-MSH mutant fish compared with the WT controls. The growth rate was determined by energy absorption and conversion efficiency, as well as the individual metabolic rate.

We selected appetite regulatory genes that have been demonstrated to take part in regulating food intake and to serve as indicators of controlling the anorexigenic and orexigenic functions and compared their expression patterns during preprandial stages between the pomc mutant fish and WT control. We observed that the anorexigenic genes (*sim1a*, *crhb*, *trh*, and *bdnf*) were significantly downregulated in the α-MSH mutant fish at 1.5 h after feeding. To further confirm the relationship between α-MSH and appetite regulation, we used intracranial administration of an α-MSH analog and MTII. During 1.5 h after administration, we discovered that fluorescence intensities were repressed significantly in the hypothalamus of both WT and α-MSH mutant fish. Moreover, the *bdnf* and *sim1a* expression levels in the α-MSH^−/−^ and α-MSH analog/MTII administration groups were significantly enhanced compared with the WT and saline administration control groups. Lack of α-MSH leads to suppression of anorexigenic genes expression (*bdnf*, *sim1a*, *crhb*, and *trh*) results in a reversible imbalance between food intake and energy homeostasis in α-MSH mutant fish (Figure 7A). Most of these associated features might be regulated by α-MSH [19,45], suggesting their critical functions in controlling metabolism observed in our α-MSH mutant zebrafish (Figure 7B). In line with previous reports, most metabolic defects observed in fish [62], mammalian models [4,63,64], and humans [65,66,67] have been attributed to the impairment of functional MC4R signaling in various *pomc* mutants.

## 5. Conclusions

Together with the above findings in our study, the BW gain (or obesity) and increase in body lipid contents were associated with increased appetite and decreased energy expenditure rates in this zebrafish model. In summary, we characterized a critical role for melanocortin signaling for appetite suppression. Therefore, our findings may help clarify underlying mechanisms between α-MSH signaling and obesity.

## Figures and Tables

**Figure 1 biomedicines-09-00941-f001:**
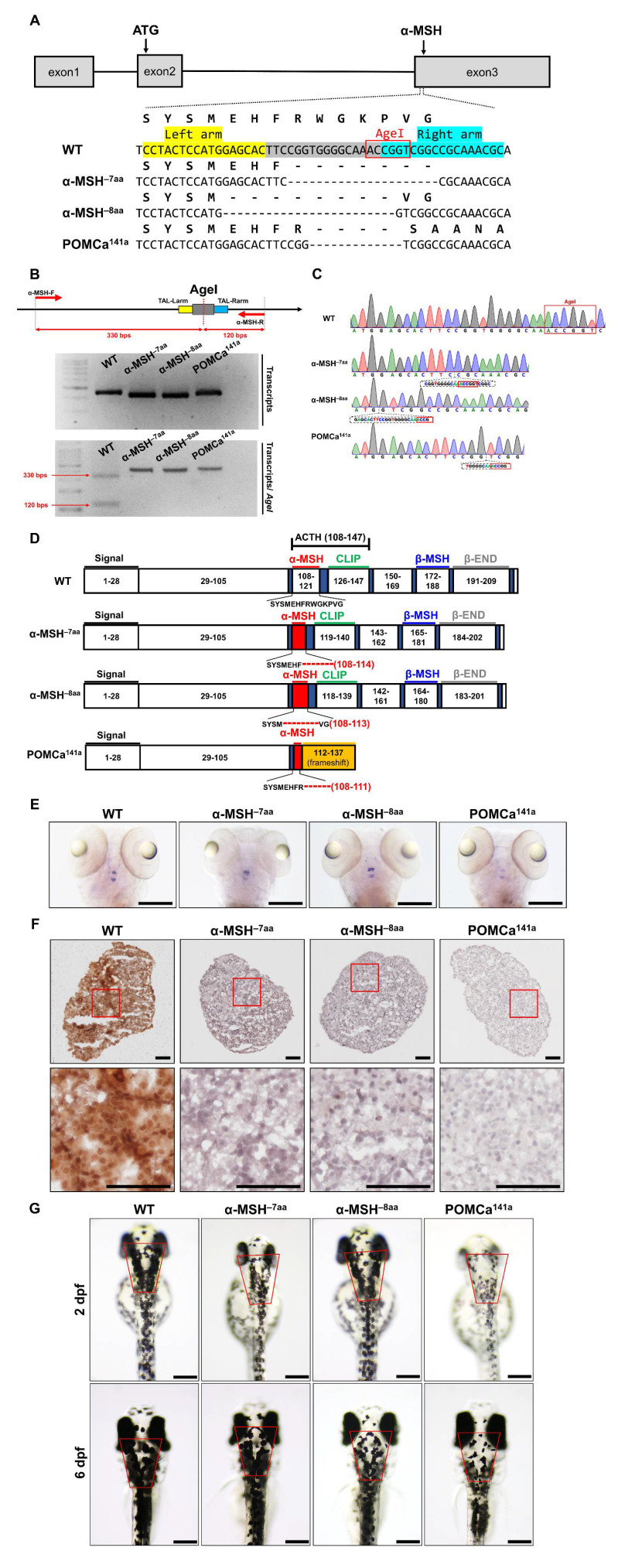
Generation of zebrafish *pomca* mutants using TALEN. (**A**) Schematics of the three *pomca* mutant alleles generated. The sequences in the third exon that were targeted by the TALEN pairs are shown in gray boxes. The AgeI restriction enzyme recognition site for genotyping purposes is shown in a red box; dashed lines indicate deleted nucleotides. The left and right TALEN targeting sites are highlighted in yellow and sky blue, respectively. (**B**) Upper: Schematic illustration of the primers used for RT–qPCR detection of mutations. The specific primer is to the mutated site/scheme of the locations of primers (red arrows) designed to detect a disruption in the spacer of the third exon. Yellow box, left TALEN sites (TAL-L); sky blue box, right TALEN sites (TAL-R); grey box, spacer; red dashed line, AgeI restriction site. Lower: Results of RT–qPCR analyses on fin clips of heterozygous F1 fish containing one of the corresponding mutations (as indicated in A). (**C**) Chromographs illustrating the sequences in the third exon of the *pomca* WT controls and the nucleotide deletion of *pomca* mutants. The boxed sequence in red indicates the restriction enzyme AgeI cutting sites in WT controls that were deleted in the mutant fish. (**D**) These diagrams show the predicated Pomca protein of *pomca* mutants compared with the WT Pomca protein. (**E**) Whole-mount ISH showing the expression of pomca transcripts in the pituitary in WT controls and *pomca* mutants larvae at 5 dpf. Scale bars = 200 μm. (**F**) Expression patterns of α-MSH in 12 mpf Pomc neuron samples after Immunohistochemistry-frozen section (IHC-F) staining. Scale bar = 50 μm. (**G**) Dorsal view of 2 dpf and 6 dpf *pomca* mutant larvae. Scale bars = 200 μm.

**Figure 2 biomedicines-09-00941-f002:**
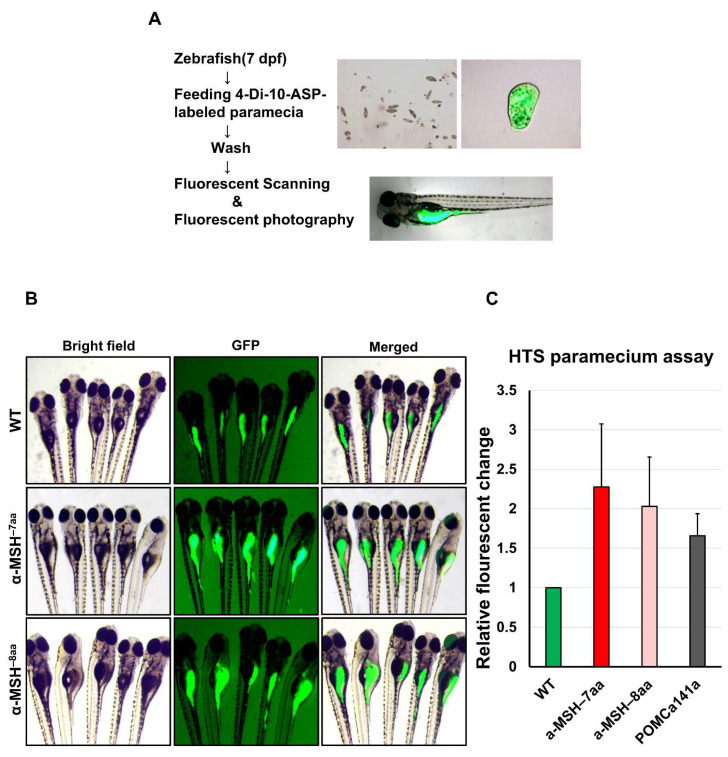
A qualitative food intake assay for zebrafish *pomca* mutant larvae. (**A**) Schematic representation of the feeding assay. Fluorescent intensities after free-feeding of 4-10-Di-ASP-labeled paramecia. Fluorescent intensities of ingested paramecia were at maximum levels 1.5 h after feeding (see Methods). (**B**) Side views of 7 dpf larvae examined under fluorescence illumination. The larvae were incubated with fluorescent microspheres coated with fish food for up to 1.5 h before visualizing the fluorescent contents in their gut. (**C**) Correlation between the relative amount of paramecia and fluorescent intensities of ingested paramecia in zebrafish. The fluorescent intensities were measured from the numbers of introduced paramecia thrice independently. All values are the mean ± SEM, *n* = 50.

**Figure 3 biomedicines-09-00941-f003:**
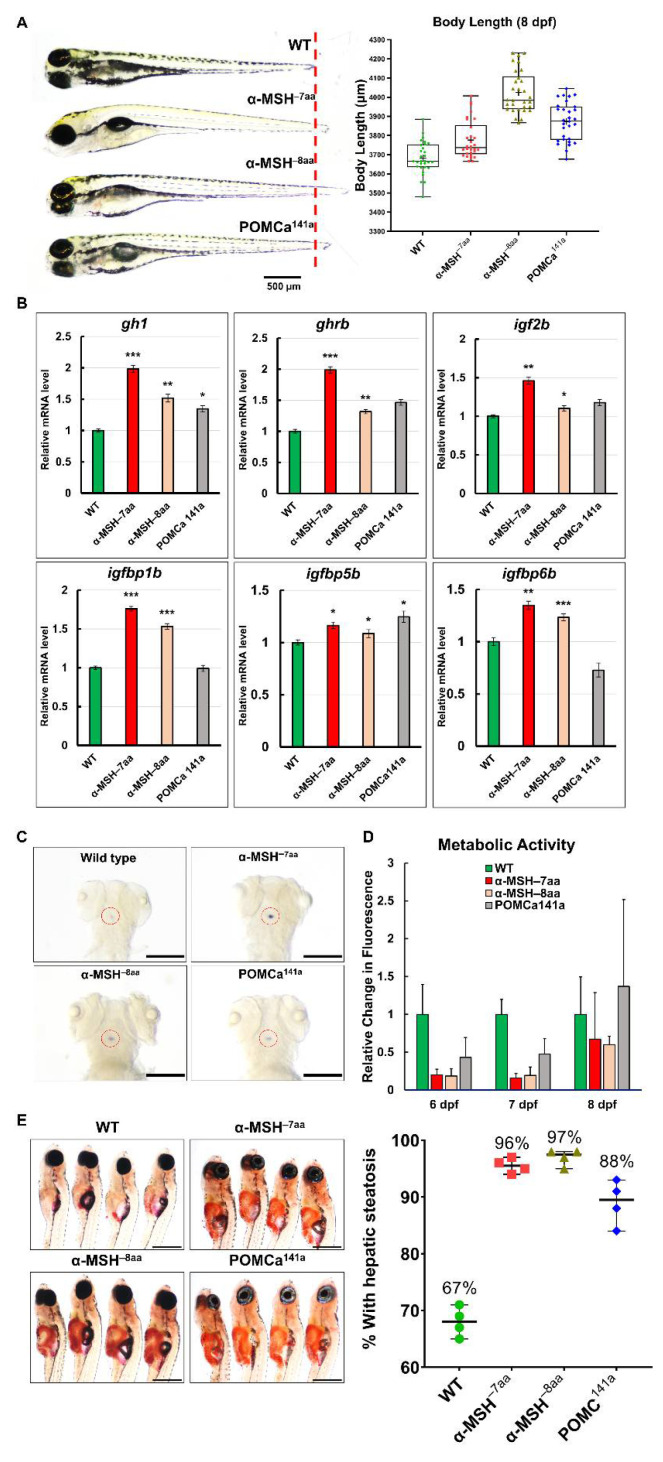
The level of α-MSH regulates normal somatic growth and energy balance in zebrafish embryos/larvae. (**A**) Left: Lateral view of wild-type and *pomca* mutant larvae at 8 dpf. Scale bar = 500 μm. Right: Statistical analysis of BL (jaw to tail fin) in WT controls and *pomca* mutant larvae at 8 dpf. Data are shown as the mean ± SD (*n* = 30). (**B**) Expression of GH/IGF axis genes, *gh1*, *ghrb*, *igf2b*, *igfbp1b*, *igfbp15b*, and *igfbp6b*, in WT controls and *pomca* mutants larvae at 8 dpf. Values are the mean ± SEM. * *p* < 0.05, ** *p* < 0.01, and *** *p* < 0.001 compared with WT groups. (**C**) Whole-mount ISH showing the increased expression of *gh1* in the pituitary in *pomca* mutants larvae at 5 dpf. Scale bars = 200 μm. (**D**) Response to depletion of α-MSH in *pomca* mutant larvae by the Alamar Blue assay. Three larvae per well were incubated in 4 mM sodium bicarbonate with 1% Alamar Blue. The fluorescence of the solution was measured at different time points. Data are reported as the relative change in fluorescence intensity at least three times independently. (mean ± SEM, *n* = 30). (**E**) Left: Hepatic steatosis was observed by whole-body Oil Red O staining in *pomca* mutant larvae at 21dpf. Right: Percentages of WT and *pomca* mutant larvae with strong levels of hepatic steatosis at 21 dpf were calculated from at least 50 fish in each group. Data were representative of four independent experiments. Scale bar = 1 mm.

**Figure 4 biomedicines-09-00941-f004:**
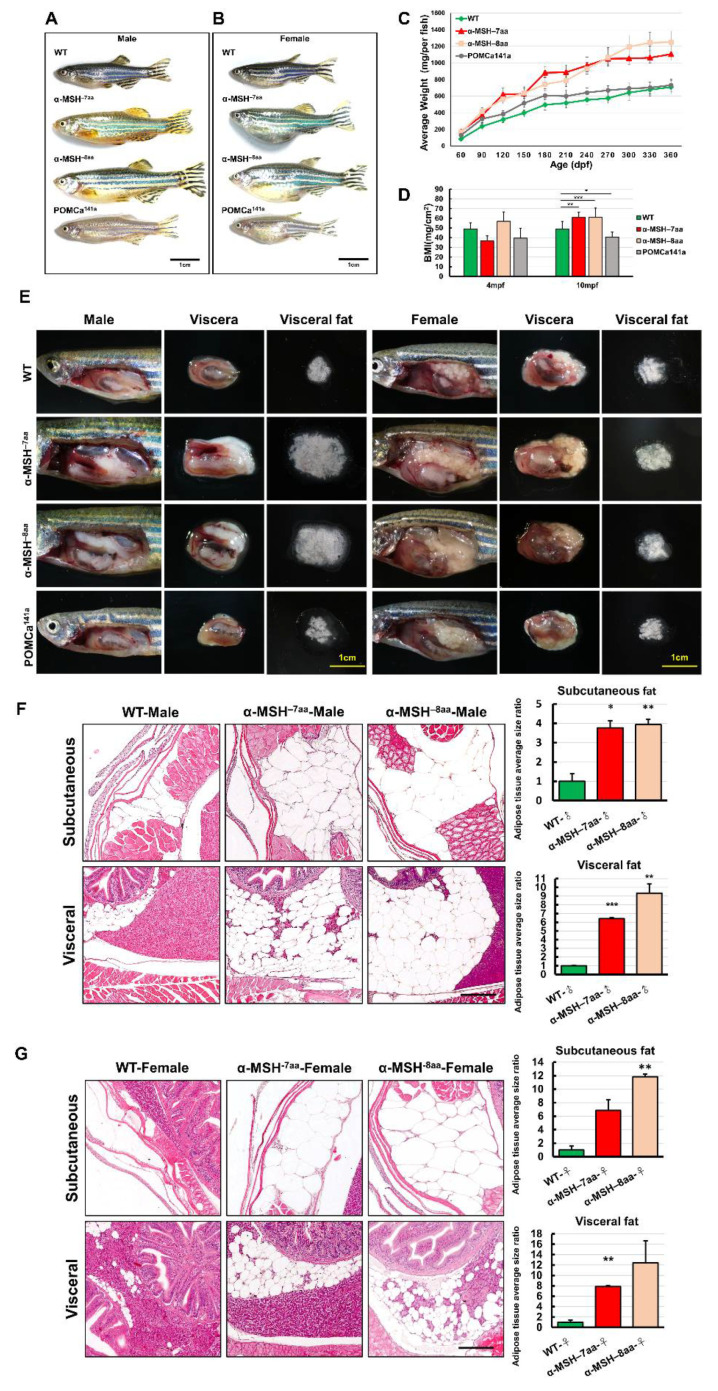
The level of α-MSH regulates somatic growth and sexual size dimorphism in *pomca* mutant adults. (**A**,**B**) Lateral view of F2 homozygous male (**A**) and female (**B**) mutants compared with WT controls at 12 mpf. Scale bars = 1 cm. (**C**) BW curves of zebrafish from the juvenile to adult stages in the three experimental male groups (WT, α-MSH^−7aa^, and α-MSH^−8aa^; *n* = 20/group). (**D**) Bar graph showing the BWs in the three experimental male groups (*n* = 20/group) under normal feeding conditions at 4 and 10 mpf stages. (**E**) Whole mounts of viscera and visceral fat pads in the α-MSH^−7aa^ and α-MSH^−8aa^ fish groups at 12 mpf, showing increased fat pad size in the α-MSH. (**F**,**G**) Histological features of adipose tissue in hematoxylin and eosin (HE)-stained sagittal sections, showing (**F**) male and (**G**) female specimens. Left: HE-stained sagittal sections, showing visceral and subcutaneous adipose tissue contents in the three experimental male groups (*n* = 5/group). Right: Bar graph showing the body fat volume ratios calculated by morphometric analysis of fat on visceral and subcutaneous adipose tissue average size in each experimental group (*n* = 5 in each group). Values are the mean ± SEM. * *p* < 0.05, ** *p* < 0.01, and *** *p* < 0.001 compared with WT control groups.

**Figure 5 biomedicines-09-00941-f005:**
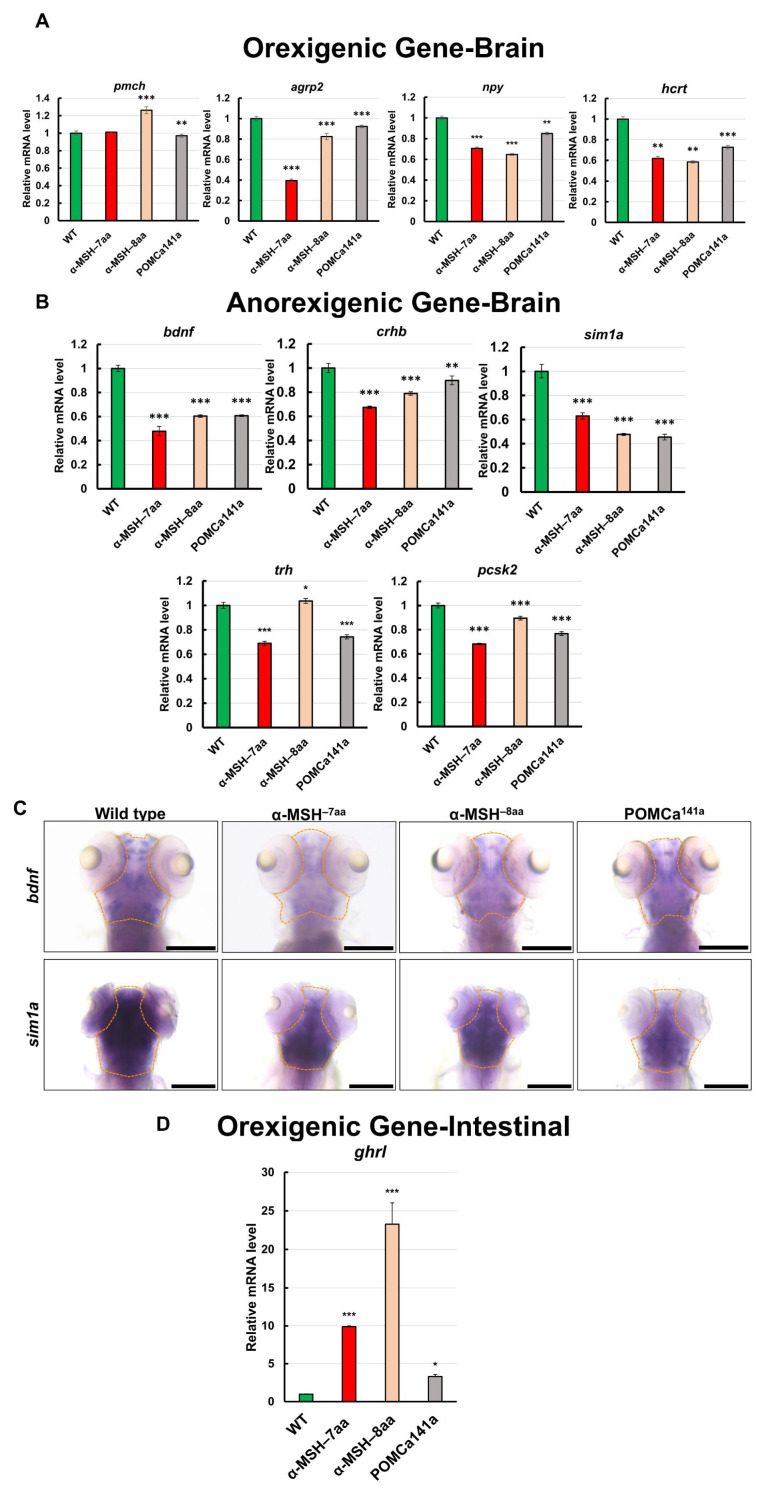
Effects of α-MSH on feeding regulation in the hypothalamus of zebrafish. (**A**,**B**) RT–qPCR analysis was used to measure the mRNA levels of four selected orexigenic genes, *pmch*, *agrp2*, *npy*, and *hcrt* (**A**), and five selected anorexigenic genes, *bdnf*, *sim1a*, *crhb*, *trh*, and *pcsk2* (**B**) in the WT and *pomca* mutants at 1.5 h after feeding. (**C**) Whole-mount ISH showing decreased expression of *bdnf* and *sim1a* in the pituitary in WT and *pomca* mutant larvae at 5 dpf. Scale bars, 200 μm. (**D**) RT–qPCR analysis was used to measure the mRNA level of the intestinal orexigenic gene, *ghrl*, in WT and *pomca* mutant adults. Values are means ± SEM. * *p* < 0.05, ** *p* < 0.01, and *** *p* < 0.001 compared with WT control groups.

**Figure 6 biomedicines-09-00941-f006:**
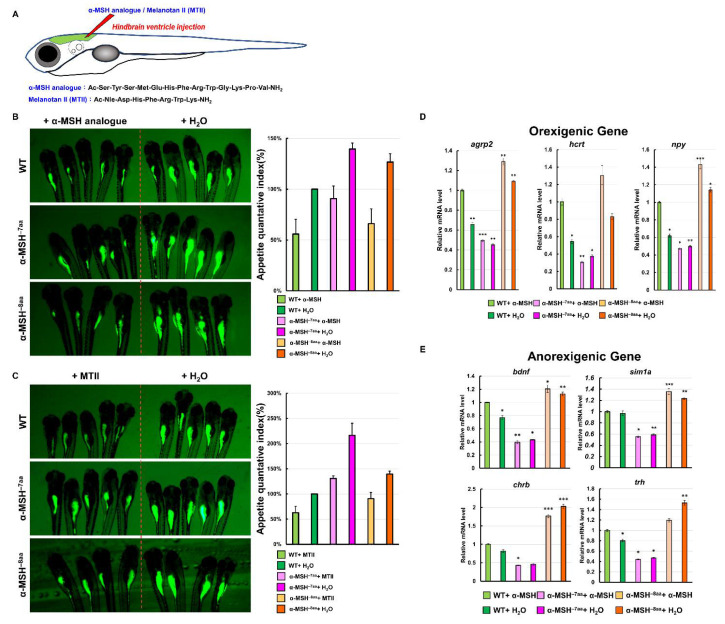
Effects of a synthetic α-MSH analog in rescuing hyperphagic phenotypes in α-MSH mutant larvae. (**A**) Schematic representation of hindbrain ventricle injection of an α-MSH analog/MTII into α-MSH mutant larvae. (**B**) α-MSH analog, and (**C**) MTII. Left: Administration decreased feeding volume in α-MSH mutant larvae at 7 dpf. Right: Bar graph showing the quantified appetite levels measured by morphometric analysis of fluorescent intensities in each experimental group (WT, α-MSH^−7aa^, and α-MSH^−8aa^) (*n* = 50/group). Values are means ± SEM. * *p* < 0.05, ** *p* < 0.01, and *** *p* < 0.001 compared with WT groups. (**D**,**E**) RT–qPCR analysis was used to measure the mRNA levels of three selected orexigenic genes, *agrp2*, *npy*, and *hcrt* (**D**), and four selected anorexigenic genes, *bdnf*, *sim1a*, *crhb*, and *trh* (**E**) in the WT and α-MSH mutant larvae at 1.5 h after feeding. We used relative fluorescent change and relative transcriptome expression for quantification three times (*n* = 50) independently.

**Figure 7 biomedicines-09-00941-f007:**
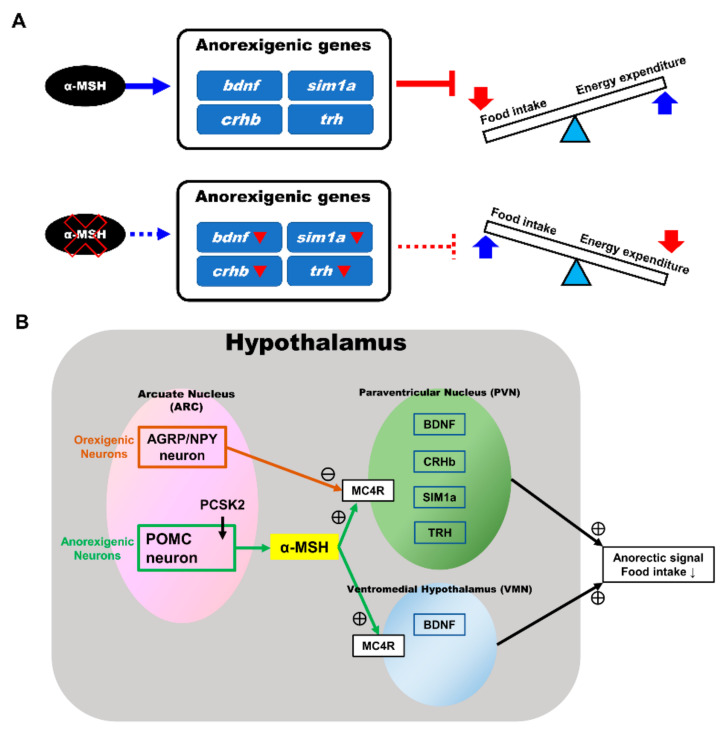
Anorexigenic signals in hypothalamic neural circuits promoted by α-MSH expression in zebrafish. (**A**) A proposed mechanism that α-MSH mediating food intake and energy expenditure via anorexigenic molecules manipulation which can reversed by α-MSH depletion. (**B**) Schematic presentation of anorexigenic factors (*bdnf*, *sim1a*, *crhb*, and *trh*) enhancement within the paraventricular nucleus (PVN) and the ventromedial hypothalamus (VMN) by α-MSH resulting in a dramatic outcome with the anorexigenic phenomenon.

## Data Availability

The data presented in this study are available upon request from the corresponding author.

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
