# Peer review of "Depletion of Alpha-Melanocyte-Stimulating Hormone Induces Insatiable Appetite and Gains in Energy Reserves and Body Weight in Zebrafish"

_biomedicines, 2021, doi:10.3390/biomedicines9080941_

Round 1

Reviewer 1 Report

The manuscript reports the effects of alpha-MSH knockout in appetite-regulating Pomc neurons in zebrafish, corroborating experiments in other species and validating the zebrafish model of genes regulating appetite. The experiments are comprehensive and appear very thorough, making this contribution to the literature on appetite control appropriate and timely, if not novel or new. The main omission in the manuscript is the absence of sample sizes of the various experiments. Only twice, 20 fish per diet in the growth rate/BMI experiment (Fig. 4), stated in the methods and figure caption (also stating n=5 for the fat content assessment), and 50 fish/treatment stated in the caption of the a-MSH rescue experiments of Fig. 6 (restated in caption of Fig 2 also on these same fish) is this unambiguously stated. Meanwhile, some of the experimental means appear extremely close, with tiny standard errors of the mean, yet significant differences via ANOVA with post-hoc Bonferroni are claimed. This implies that n was very large, but in any case the sample sizes must be reported, independent of references to other published papers by the group. When combined with the very low resolution of the figures, these omissions make verification of differences difficult. This is the main issue that must be addressed for the results to be independently verified. Low resolution of figures in this manuscript make it hard to corroborate results. I am not holding the authors fully to blame for this (making the figs larger to begin with may have helped, which is something the authors likely have control over), and please note I have brought it up with the editor to correct for authors in the future. However, reviewing the paper necessitates studying the figures and drawing independent conclusions about the data, which is difficult when they are so difficult to read and study. Fig. 3A right: the resolution of this fig makes it impossible to distinguish *, ** and *** even at a view of 200% in Adobe Acrobat Pro. Similar low res makes it impossible to confirm in Fig. 3C caption (and lines 257-9) that the pomca mutant has increased gh1 expression. Fig. 4A by now I know what the mutants are and the sequence in which they are presented in figs. But the labels on the mutants are blurry due to low res, and Fig. 4C, 4D the keys to these figs are unreadable due to low res Fig. 4F, G histograms, same. It is difficult to resolve white space as fat droplets in the histological sections of Figs. 4F, G. Fig. 5A: are pmch and agrp2 in pomca mutants truly significantly different from WT? each mean, especially in pmch, looks nearly identical to WT. See earlier discussion of absence of sample sizes. Several grammatical and stylistic changes are requested: Lines 84-85: disturb …obesity … Please correct this phrase. Line 171: gender fish don’t think of themselves as male or female, the anthropological definition of gender, so please substitute the word: sex Line 297 say significantly different … not statistically significant

Reviewer 2 Report

In this manuscript the effect of a-MSH depletion is studied. The results show that a-MSH has a role as an appetite regulator.

The structure of the work is very well planned and organized. The results presented are clear and the diagrams, for example figure 7) clearly summarize the results.

This manuscript is of very good quality, and I only have a few suggestions / comments that could improve it:

  • The authors have used a-MSH antibody (1: 1000, RayBiotech 130-10355). They should also indicate in section “2.7. Histology and Immunohistochemistry (IHC) ” the secondary antibody, concentration and incubation protocol.
  • In section “2.4. In-Situ Hybridization (ISH) ” the authors refer to a previous work for the description of the protocol (“ ISH was performed as described previously [34] ”), however, in paper 34, they cite another, and in that 2 others, therefore, it is not easy to find the original article describing the method, which I think is Jowett et al., 2001 (METHODS 23, 345–358). In my opinion, the authors should cite the original work that describes the method.
  • Why in figure 1F, which describes the "expression patterns of a-MSH in 12 mpf Pomc neuron samples after Immunohistochemistry-frozen section", the color of the WT is different from the other groups?
  • In some figures, for example 2c or 3d, e, bar graphs are shown. I suggest using swarm plots. Swarm plots are clearer than bar graphs because they show all the observed points, it clearly shows the number of samples, the value of each sample and the variability, therefore changing the graphs to a swarm plot would improve the clarity which the results are presented.

I hope that my comments help to improve the quality of the manuscript.
